# CVD and COVID-19: Emerging Roles of Cardiac Fibroblasts and Myofibroblasts

**DOI:** 10.3390/cells11081316

**Published:** 2022-04-13

**Authors:** Laxmansa C. Katwa, Chelsea Mendoza, Madison Clements

**Affiliations:** Department of Physiology, Brody School of Medicine, East Carolina University, Greenville, NC 27834, USA; chelseamendoza86@gmail.com (C.M.); madisonaclements@gmail.com (M.C.)

**Keywords:** TGF-β1, fibroblast, myofibroblast, fibrosis, COVID-19

## Abstract

Cardiovascular disease (CVD) is the leading cause of death worldwide. Current data suggest that patients with cardiovascular diseases experience more serious complications with coronavirus disease-19 (COVID-19) than those without CVD. In addition, severe COVID-19 appears to cause acute cardiac injury, as well as long-term adverse remodeling of heart tissue. Cardiac fibroblasts and myofibroblasts, being crucial in response to injury, may play a pivotal role in both contributing to and healing COVID-19-induced cardiac injury. The role of cardiac myofibroblasts in cardiac fibrosis has been well-established in the literature for decades. However, with the emergence of the novel coronavirus SARS-CoV-2, new cardiac complications are arising. Bursts of inflammatory cytokines and upregulation of TGF-β1 and angiotensin (AngII) are common in severe COVID-19 patients. Cytokines, TGF-β1, and Ang II can induce cardiac fibroblast differentiation, potentially leading to fibrosis. This review details the key information concerning the role of cardiac myofibroblasts in CVD and COVID-19 complications. Additionally, new factors including controlling ACE2 expression and microRNA regulation are explored as promising treatments for both COVID-19 and CVD. Further understanding of this topic may provide insight into the long-term cardiac manifestations of the COVID-19 pandemic and ways to mitigate its negative effects.

## 1. Introduction

According to the World Health Organization, in 2019, approximately 17.9 million individuals died due to cardiovascular disease (CVD), accounting for 32% of global deaths [1]. The rise of coronavirus disease-19 (COVID-19) is expected to only exacerbate climbing CVD statistics due to cardiovascular injury caused by the infectious disease [2]. Many COVID-19 patients have an increase in cardiovascular damage, particularly acute cardiac injury [3,4], typically indicated by increased troponin levels [5]. In one study performed by Moody et al. [3], 79 hospitalized patients with COVID-19-induced pneumonia underwent two separate transthoracic echocardiographs (TTE), one being a baseline at a median of 8 days after admission and one being 3 months after the baseline. From the baseline TTEs, 55% of patients showed evidence of adverse ventricular remodeling. After 3 months, 29% still had evidence of adverse ventricular remodeling, indicating that cardiovascular damage caused by COVID-19 may be long-term. Moreover, patients with pre-existing CVD are at a higher risk of severe disease than those without [6]. However, due to the novel nature of SARS-CoV-2, its effects on the cardiovascular system are not well understood. This has caused great interest within fields of research and medicine alike to elucidate why SARS-CoV-2 is more fatal to patients with CVD and how it can exacerbate pre-existing cardiovascular damage. The cardiac fibroblast is a well-studied cell type responsible for maintenance and wound healing within the heart. Additionally, cardiac fibroblasts are known for their role in contributing to cardiac fibrosis and adverse remodeling. The fibroblast/myofibroblast’s properties as both healing and harming within the heart make them promising candidates for future research concerning COVID-19-related cardiac injury.

Cardiac fibroblasts are the most prevalent cells in the heart [7]. They are identified using cytoskeletal marker vimentin and cell surface marker discoidin domain receptor 2 (DDR2) [7,8]. Fibroblasts maintain the extracellular matrix (ECM) and the structure of the heart through the production of interstitial collagen [9]. In addition, cardiac fibroblasts are widely known for their response to injury, such as myocardial infarction (MI) [10]. Typically when myocardial infarction occurs, fibroblasts differentiate into an activated phenotype (myofibroblasts) through the activation of transforming growth factor-beta 1 (TGFβ-1) via angiotensin II (Ang II) [9]. Fibroblast activation may also be induced by the release of cytokines, various growth factors, or neurohumoral pathways [11]. Myofibroblasts are a specialized phenotype of cardiac fibroblasts, which express alpha-smooth muscle actin (α-SMA), vimentin, and pro-remodeling factors such as endothelin-1 (ET-1) and Ang-II [9]. This allows for the increased production of collagen and other components beneficial for wound healing. Eventually, scar formation occurs, ideally signaling the apoptosis of myofibroblasts and leaving a healed scar behind [9].

The pathological conditions that can arise when fibroblasts differentiate into myofibroblasts are well-studied. After wound healing, when myofibroblasts fail to undergo programmed cell death, their persistence in the heart may cause excessive collagen synthesis and accumulation in the myocardial interstitium, eventually resulting in adverse remodeling and fibrosis [10,12,13]. Recently, new molecular factors contributing to fibroblast-myofibroblast differentiation and fibrosis have been discovered [5,14]. This includes increased levels of Ang II in COVID-19 patients due to the initial mechanism of binding for the SARS-CoV-2 virus. Increased Ang II levels may accelerate cardiac fibroblast to myofibroblast differentiation [9]. Additionally, microRNAs (miRNAs), specifically miR-125b, seem to be upregulated in fibrotic human hearts and myofibroblasts [14]. However, because of our limited knowledge of the different processes behind fibroblast–myofibroblast differentiation, unknown mechanisms could be contributing to the accumulation of collagen, leading to cardiac dysfunction.

The aim of this review is to expand upon the current literature concerning the role of cardiac fibroblasts and myofibroblasts in adverse remodeling and cardiac fibrosis and to introduce new concerns regarding their role in COVID-19-induced cardiovascular damage. Additionally, we aim to present emerging evidence regarding potential treatment options for patients suffering from CVD and COVID-19.

## 2. Cardiac Fibroblasts in Response to Injury

The cardiac fibroblast comes as a major benefactor to the phases of inflammation, the proliferation of non-myocytes, and scar formation by creating connective tissue in the heart. In response to an injury in the myocardium, inflammation signals activate cardiac fibroblasts, leading them to emit cytokines, prostaglandins, and leukotrienes [15,16], as well as matrix metalloproteinases (MMPs) to clear the injured area of necrotic cells and debris [17]. The produced cytokines induce fibroblast to myofibroblast differentiation, which then furthers the inflammatory response [17].

In a healthy heart, fibroblast activation occurs to provide stronger pro-remodeling factors for the injury. As seen in Figure 1, while resident fibroblasts make up a majority of activated fibroblasts in the heart [9], circulating hematopoietic cells, endothelial cells, progenitor stem cells, and pericytes [10] are speculated to become myofibroblasts as well. It remains unclear if they truly do. However, if these speculated cell types do contribute to the proliferation of activated fibroblasts, they may evoke further cardiac problems [10]. Once activated, more fibroblasts and immune cells are attracted to repair the site of injury in the heart. This could be happening through the cytokines and other mediators that are released by these cells or coming from circulation [9].

Cardiac fibroblast activation unlocks a variety of phenotypic characteristics, such as the enhanced production of collagen facilitated by pro-remodeling factors, including ET-1, Ang II, and TGF-β1 [9]. Myofibroblasts increase collagen turnover to repair damaged heart tissue at the site of injury and in healthy tissue near the injury [10]. As regenerative tissue forms, the collagen’s strength increases as it comes in contact with the injured area [10]. Myofibroblasts also express alpha-smooth muscle actin (α-SMA), a contractile protein that allows the cells to assist in closing wounds [7,9]. The release of these pro-remodeling factors is used to build a stronger ECM and connect the newly created regenerative tissue. Generally, α-SMA is expressed at the site of myocardial injury. Apart from myofibroblasts, α-SMA is also expressed by vascular smooth muscle cells, which are increased in the vessels and surrounding areas after injury [18]. Sometimes this poses a problem in locating myofibroblasts, specifically in areas close to injured vessels.

After the granulation tissue is produced, myofibroblasts typically undergo apoptosis to allow the scar to mature. Failure to undergo apoptosis can result in excessive collagen production [7,9].

### Cardiac Fibrosis and Adverse Remodeling

Cardiac fibrosis is the result of excessive accumulation of fibrous ECM proteins in the heart, which progressively leads to cardiac dysfunctions and, ultimately, heart failure [12]. The most robust forms of fibrotic tissue or scar formation occur in response to acute myocardial injury, but other factors such as age, obesity, diabetes, and hypertension are also known to contribute to cardiac fibrosis [12]. When myocardial infarction occurs, a scar must be produced to prevent the rupture of the dead myocardium, as the damaged cardiomyocytes are permanently lost [19]. Although this remodeling is beneficial for repair, excessive changes to the normal ECM structure can have adverse effects [20]. This process initially increases the deposition of ECM proteins causing the loss of contractile function of the heart tissue due to myocardial stiffness and disrupted communication between the cardiomyocytes [7,9,10]. Adverse remodeling is also exacerbated by the loss of cardiomyocytes through necrosis and apoptosis [13]. As the fibrosis continues to progress, left ventricle hypertrophy and systolic dysfunction may occur, potentially giving way to heart failure [7,9]. Altogether, these factors encompass a compilation of harmful effects on cardiac structure and function at various magnitudes [13].

The most well-studied fibroblast activation pathway begins when Ang II activates TGF-β1 through the TGF-β1 signaling pathway, inducing a phenotypic switch. This signaling continues to take place, leading to the accumulation of myofibroblasts in the myocardium. TGF-β binds to the TGF-β-receptor II, which phosphorylates TGF-β receptor kinase 1 (TβR1) and activates SMAD-dependent TGF-β canonical pathway. This, in response, induces fibrosis [10]. To stop the accumulation of myofibroblasts in the heart, scientists suggest that certain signaling pathways, specifically TAK1 and p38, should be inhibited [10].

According to Ono et al. [21], TGF-β-activated kinase 1(TAK1) increases ECM protein production, contributing to the development of fibrotic disorders. To stop this, a dominant-negative TAK1 inhibitor (TAK1-DN) needs to bind with the kinase. From their experiment, TAK1-DN proved to reduce ECM protein production and has the ability to intercept TGF-β1 signaling. Additionally, in a study by Li et al. [22], when upregulating activating transcription factor-3 (ATF3) expression, MAP2K3 expression was suppressed and later inhibited p38 signaling. Inhibition of p38 reduced the expression of TGF-β signaling-related genes. ATF3 is notably a responsive gene in cardiac fibroblast activation [23]. When activated, the gene proceeds to regulate the cellular mechanisms with other ATF genes by activating or repressing working genes [23]. In this case [22], ATF3 inhibits p38 signaling making its upregulation during cardiac failure a self-protective mechanism from fibrosis and, consequently, adverse remodeling. Thus, inhibiting cell signaling pathways that induce the incident of fibrosis could come as a potential therapeutic approach for individuals with MI caused by cardiac fibrosis. Furthermore, manipulating signaling pathways has come as an insightful way to determine what mechanisms are present for cardiac fibrosis to occur.

## 3. Emerging Cardiac Conditions Relating to COVID-19

Recent literature indicates that CVD is a risk factor for severe COVID-19 and is defined as a cardiac tissue insult in response to the virus’ interaction. Further myocardial damage is likely to be exhibited if a SARS-CoV-2 patient initially had a CVD [24]. Among the 44,672 cases of COVID-19 reported by the Chinese Center for Disease Control and Prevention, the population determined a case-fatality rate of 10.5% with pre-existing comorbid CVDs [25]. Furthermore, a meta-analysis performed by Li et al. [26] identified that among the 1527 patients with COVID-19 in Wuhan, China, the prevalence of hypertension and cardiac/cerebrovascular disease was 16.7%, 17.1%, and 16.4%, respectively. Acute myocardial injury was reported in at least 8% of these patients and tended to have a higher incidence rate in ICU COVID-19 patients [26].

### 3.1. COVID-19-Induced Cytokine Storm

The infiltration of SARS-CoV-2 into respiratory epithelial cells activates an immune response characterized by the production of pro-inflammatory cytokines [27]. This includes interleukin 6 (IL-6), IL-12, IL-1β, and interferon γ [28]. The overproduction of these cytokines and chemokines causes a “cytokine storm”, is often an indicator of severe disease, and precipitates acute respiratory distress syndrome [29]. In a meta-analysis by Coomes and Haghbayan [30], including 10 studies with cytokine levels reported in COVID-19 patients, all studies found increased levels of IL-6, with five studies finding higher levels in patients with severe disease compared to those with moderate disease.

The expression of cytokines, particularly IL-1β and IL-6, is also prominent in fibrotic hearts [12]. Hypertension is a known risk factor for fibrosis, and increased plasma IL-6 levels, mediated by Ang II, tend to correspond with increased systolic blood pressure and decreased endothelial function [31], indicating that IL-6 may play a role in increasing blood pressure and decreasing cardiac function [32]. Additionally, cytokine release can induce cardiac fibroblast activation, potentially promoting fibrosis and adverse remodeling [11]. Thus, the overproduction of said cytokines and chemokines is likely contributing to the cardiac damage seen in severe COVID-19 patients.

### 3.2. ACE2’s Involvement in Viral Infection

Initially, SARS-CoV-2 binds to ACE2 receptors notably expressed in the lungs, heart, and vessels but found throughout the body as well. In a study published by Zou et al. [33], the proportion of cells expressing ACE2 in several tissues throughout the body was calculated. It was determined that tissues with >1% ACE2 expression were at high risk for viral infection. The tissues which fit this distinction are as follows: ileum (30%), heart (>7.5%), kidney (4%), bladder (2.4%), respiratory tract (2%), and esophagus (>1%) [33]. The high proportion of ACE2 expressing cells in the heart may explain why severe myocardial injury is often a symptom of severe COVID-19.

When SARS-CoV-2 interacts with ACE2, the enzyme loses contact with the cell surface, causing Ang II levels to increase and Ang-(1–7) levels to decrease, as seen in Figure 2 [5,34]. Recent studies have demonstrated elevated levels of autoantibodies targeting angiotensin II type 1 receptor (AT1R) in patients with moderate to severe COVID-19 compared to milder cases [35] and versus healthy controls [36]. This indicates an immune response similar to that of autoimmune disease. Activation of AT1R, typically via elevated Ang II expression, can thus induce cardiac fibroblast differentiation activity and ultimately lead to the proliferation of myofibroblasts [9]. Hence, this mechanism may explain why patients with COVID-19 and underlying cardiac conditions show vascular inflammation, endothelial dysfunction, and thrombosis. Since ACE2 is widespread in the body, increased Ang II levels may also occur in organs other than the heart and create inflammation in more advanced COVID-19 patients [5]. In addition, the renin–angiotensin–aldosterone system (RAAS) is affected by sex, as is the expression of ACE2. In male cardiac cells, levels of soluble angiotensin-converting enzyme 2 (sACE2) tend to be increased [37]. It has been theorized that increased levels of ACE2 could explain why males tend to have worse COVID-19 outcomes and increased mortality compared to females [38].

Over the past year, it has been noted that the mortality rate for individuals with underlying chronic diseases, specifically cardiometabolic diseases, has surged compared to the time before the pandemic [39]. A wide range of cardiac complications is present in patients with COVID-19 that have significantly worsened their health, including myocardial injury, myocarditis, acute myocardial injury, and heart failure [40]. Myocardial infiltration of immune cells, activation of myofibroblasts, and cardiomyocyte necrosis are a few of the short-term complications of COVID-19 that could lead to inflammation; local and systemic inflammation plays a significant role in further evoking existing cardiac conditions. Long-term effects that could be caused by inflammation include cardiac fibrosis, cardiac hypertrophy, and decreased cardiac output [41], as well as ischemic myocardial injury with its other related mechanisms such as plaque disruption, coronary spasm, and microthrombi. These conditions could eventually lead to cardiac dysfunction, observed as responses to inflammation and, possibly, direct viral action [5].

### 3.3. MicroRNAs

MicroRNAs (miRNAs) are small non-coding RNA molecules with the ability to negatively regulate gene expression [14,42]. MiRNAs are factors that can both contribute naturally to cardiac fibrosis and can help prevent it depending upon which genes they regulate [43]. MiR-125b is one example of miRNAs that are considered major factors in the induced transition from cardiac fibroblasts to myofibroblasts [14]. According to Nagpal et al., this microRNA is significantly enhanced in the TGF-β1 induced EndMT by Ang II once it is activated by cardiac stress. Ang II induces differentially expressed microRNAs in cardiac fibroblasts [44]; this includes miR-125b. Once this microRNA is expressed, it can suppress factors such as p53 and apelin that regulate cardiac fibroblast-myofibroblast differentiation. Furthermore, miR-125b’s induction can collectively affect TGF-β pathways and cause mutations in regulation mechanisms to promote the proliferation of myofibroblasts. Overall, miR-125b can repress anti-fibrotic mechanisms, alter gene expression of important fibrotic-related genes, and inhibit regulator proteins that induce fibroblast proliferation. This miRNA has a significant effect on the regulation of fibroblast-myofibroblast differentiation and can lead to detrimental fibrotic disorders [14]. Alternatively, other miRNAs have the ability to regulate cardiac fibrosis. MiR-101a has been shown to alleviate cardiac fibrosis after myocardial infarction by decreasing ECM deposition in mouse models [45].

In a study by Lu et al. [42], miRNAs are believed to have the ability to target ACE2 receptors and prevent severe SARS-CoV-2 infection. As discussed previously, the ACE2 receptor serves as the point of entry for SARS-CoV-2 [5]. Their data suggest that miR-200c is a considerable factor in downregulating the expression of ACE2 in both rodent and human cardiomyocytes. MiR-200c’s upregulation helps repress the expression of ACE2, thus potentially making cells less susceptible to infection. MiRNAs are beneficially used to target the SARS-CoV-2 receptor and could be used as potential therapies for patients with COVID-19 and underlying conditions. However, the usage of miR-200c in patients with underlying CVD would need to be closely monitored, as many patients with CVD are treated with medication such as ACE inhibitors that leads to increased ACE2 expression [42]. Therefore, it is unknown whether the benefits of miRNA-based therapies outweigh the potential negatives; further research is required.

### 3.4. TGF-β1’s Role in the Immune Systems of Severe COVID-19 Patients

As discussed previously, TGF-β1 serves many roles throughout the cardiac system and is also an important cytokine in regulating inflammation and immune response [46]. Recent research suggests that TGF-β1 may also play a role in worsening severe SARS-CoV-2 infection via suppression of immune response [46,47].

TGF-β1 is known to suppress the functioning of natural killer (NK) cells. NK cells are innate lymphocytes that function to kill virally infected cells and prevent viral spread throughout the body. While NK cells’ role in fighting COVID-19 specifically is not known, they typically are effective in limiting the spread of RNA viruses in the body. In a recently conducted study by Witkowski et al. [47], patients with severe COVID-19 and relatively normal NK cell counts showed more expedient SARS-CoV-2 viral load decline than patients with lower NK cell counts. This suggests that NK cells do, in fact, play a major role in controlling viral spread in COVID-19 cases. This study also found that hospitalized COVID-19 patients had an earlier peak in serum TGF-β1 levels than less severe ambulatory patients. TGF-β1 may have suppressed NK cell functioning in these patients too soon, resulting in inadequate immune response [47].

Increased TGF-β1 expression is now a hallmark of severe COVID-19 cases [46,47], and it has long been a well-known fibrotic marker [9]. Consequently, severe COVID-19 cases seem to see increased incidences of adverse cardiac remodeling in both the right and left ventricles [3]. It is possible that increased TGF-β1 expression in such cases induces cardiac fibroblast to myofibroblast differentiation, resulting in an increased incidence of adverse remodeling. This possibility should be researched further to determine whether this connection is significant.

### 3.5. Myocarditis Following mRNA COVID-19 Vaccination

Myocarditis is defined as inflammation in the muscle tissue of the heart [48,49]. It is characterized by chest pain, elevated troponin levels, and abnormal MRI or ECG findings [50]. Historically, viral infection has been a major cause of myocarditis, including the hepatitis C virus, coxsackievirus B, adenovirus, and parvovirus B19. Due to viral infection causing many incidences of myocarditis, anti-viral medications are typically used as treatments [48].

Recently, a growing concern has mounted regarding reports of myocarditis following mRNA COVID-19 vaccination [50]. From December 2020 to June 2021, 1226 reports of myocarditis following mRNA COVID-19 vaccination were reported within the Vaccine Adverse Reporting System (VAERS) [49]. Among the 1212 cases with sex reported, 76.2% were male (923), and 23.8% were female (289). Among the 1194 cases with patient age reported, 57.5% (687) were less than 30 years old. This supports the current observations that myocarditis following COVID-19 vaccination tends to occur more frequently in young men; however, VAERS is a passive reporting system, and reports are not independently reviewed by the Centers for Disease Control (CDC), so it cannot be verified that all reported cases meet CDC criteria for myocarditis [49,51].

Additionally, over the past year, numerous case studies have been published documenting myocarditis in adolescents following mRNA COVID-19 vaccination. In a study published by Marshal et al. [52], seven adolescent males aged 14–19 presented to the hospital with symptoms of acute myocarditis, including chest pain, within 4 days of receiving their second dose of the Pfizer-BioNTech COVID-19 vaccination. Upon admission, all patients tested negative for COVID-19 following real-time reverse transcriptase-polymerase chain reaction tests (RT-PCR). In addition, the criteria for multisystem inflammatory syndrome in children were not met by any of the patients. All patients were diagnosed with characteristics of myocarditis or myopericarditis following cardiac MRIs, including late gadolinium enhancement and elevated troponin levels. Following treatment, all patients saw a rapid improvement in symptoms and were released from the hospital [52]. In another study conducted by Dionne et al. [53], 15 patients between the ages of 12 and 18 presented to the hospital with symptoms of myocarditis, including chest pain and elevated troponin levels. A majority of the patients were male (14 out of 15). Of the 15 patients, all but one had received their second dose of the Pfizer-BioNTech COVID-19 vaccination within the previous 6 days. All patients were discharged from the hospital within 5 days [53].

These studies support current observations that mRNA COVID-19 vaccinations can coincide with myocarditis in a small subset of people, particularly adolescent males [49]. Despite this correlation, in June 2021, the Advisory Committee on Immunization Practices in the U.S. deemed mRNA COVID-19 vaccinations safe, as the “benefits of using mRNA COVID-19 vaccines clearly outweigh the risks in all populations…” [49].

As new variants of SARS-CoV-2 emerge, the efficacy of existing mRNA vaccines comes into question. The Delta variant in particular is highly infectious due to mutations in the receptor-binding domain (RBD), allowing the virus to bind more easily to ACE2 receptors [54]. The Delta variant contains mutations in three antigenic regions of the spike protein, which may cause decreased vaccine acquired immunity [54]. The Omicron variant has recently emerged and was declared a variant of concern by the World Health Organization (WHO) on 26 November 2021 [55]. The Omicron variant is highly mutated, with 32 mutations being found in the spike protein, 15 of which occur at the RBD [56]. In a study published by Zhang et al. [56], in serum obtained from patients with naturally acquired COVID-19 immunity, the mean neutralization ED50 against the Omicron variant decreased 8.4-fold compared to the reference strain [56]. This indicates that the Omicron variant may evade naturally acquired immune protection.

## 4. Potential Treatments and Conclusions

A large portion of the world’s population is greatly affected by CVD, resulting in an increasingly high mortality rate over the past decade [2]. Thanks to current research, various mechanisms are understood regarding the causes of CVD, such as cardiac fibrosis and adverse remodeling. The roles of cardiac fibroblasts, myofibroblasts, and miRNAs are a few of the many factors that contribute to the cardiac complications of MI patients. Recently, the COVID-19 pandemic has led researchers to believe that CVD statistics may continue to rise [2] and may cause adverse cardiac remodeling to occur [3]. COVID-19 seems to increase the production of fibrotic markers such as TGF-β1 and Ang II [5,47], as well as increase the production of autoantibodies targeting AT1R, which further dysregulate the RAAS system [35,36]. These fibrotic markers influence the functioning of cell types, including cardiac fibroblasts and myofibroblasts, potentially leading to increased proliferation of myofibroblasts. However, these emerging conditions are likely also responses to viral actions and inflammation. Normally, inflammation in an injured area leads to the activation of various growth factors and cytokines for remodeling and repair to occur. Due to COVID-19, however, this could be compromised. The heart is a complex organ, and many factors, such as blood pressure and artery sizes, affect the wound healing process. Hence, the heart’s complexity has prevented research from occurring on humans as of now, as mice are the primary animal model used.

Current approaches have been used to protect and promote regeneration in the heart. Angiotensin-converting enzyme (ACE) inhibitors, mineralocorticoids receptor antagonists, angiotensin receptor-neprilysin inhibitors, and β-blockers are pharmacological solutions that have slowed or reversed the progression of adverse cardiac remodeling. However, with the rise of the COVID-19 pandemic, it is uncertain whether these drugs will do more harm than good. ACE inhibitors and angiotensin-converting enzyme inhibitors have been found to increase ACE2 expression, possibly leaving cardiac cells more susceptible to SARS-CoV-2 infection [42]. Further research regarding ACE2 expression is necessary. Additionally, understanding the molecular mechanisms of regeneration in the neonatal mammalian heart, specifically the EndMT, could also essentially provide insight into the regenerative capacity of the adult myocardium [57,58].

Through the information collected in recent literature, a few approaches are suggested that can potentially be utilized to stop further development of cardiac fibrosis and adverse cardiac complications. Inhibiting miR-125b may be useful in treating cardiac fibrosis and other fibrotic diseases. With its ability to inhibit regulatory proteins that induce fibroblast proliferation, suppressing this miRNA can keep the differentiation of fibroblasts regulated. Upregulation of miR-101a may also be useful in reversing cardiac fibrosis by downregulating TGF-β signaling [43,45]. Other miRNAs, particularly miR-200c, can target and downregulate the expression of the ACE2 receptor [42]; therefore, treatment with those specific miRNAs could be used to potentially prevent infection. However, as noted previously, decreased ACE2 expression can induce cardiac fibrosis via the upregulation of Ang II [9]. With that in mind, using miRNAs as treatments for COVID-19 or CVD should be further researched.

Additionally, genetically engineering the phenotypic switch of myofibroblasts to myocytes is also a feasible approach to prevent fibrosis, as myocytes are essential, highly energetic cardiac contractile cells that make up the heart and its function. Genetically engineering or designing cell death intracellular approaches can also be performed to avoid the further proliferation of myofibroblasts. Research trials should be implemented to potentially use these approaches in patients with fibrotic heart (cardiac hypertrophy, myocardial infarction) and adverse remodeling. It has been concluded that cardiac fibrosis and adverse remodeling are conditions that cannot be easily cured [10,12,13]. However, there are approaches to slowing down its process [57,59]. Nevertheless, further research is needed to understand the certain mechanisms that can potentially stop complications caused by defective and overactive cardiac fibroblasts and myofibroblasts, including fibrosis and adverse remodeling.

Overall, the rise of the COVID-19 pandemic has only exacerbated prominent cardiovascular diseases such as fibrosis and adverse remodeling. Patients with pre-existing CVD are known to experience more severe COVID-19 than those without [6], and many patients also suffer acute myocardial injury as a result of the disease [3,4]. It has been well-established in the literature that cardiac fibroblasts and myofibroblasts play vital roles in response to cardiac injury, but they are also responsible for fibrosis and adverse remodeling [9,10]. Fibroblasts are activated by Ang II, TGF-β, and cytokines [9,11], all of which are upregulated during SARS-CoV-2 infection [5,27,46]. Therefore, it is not unlikely that the long-term results of this pandemic are an increase in fibrosis and adverse remodeling statistics. Additionally, although vaccines are the best method currently available to combat this pandemic, they too can cause cardiac complications in the form of myocarditis [50,52,53]. However, the risk of myocarditis from mRNA vaccination is thought to be far outweighed by the risks of contracting COVID-19 without any prior immunity [49]. Ultimately, little is known regarding the long-term cardiac implications of the COVID-19 pandemic. Further research is needed to discover the mechanisms of cardiac injury caused by SARS-CoV-2 infection and potential treatment options for people suffering from both acute cardiac injury and cardiac fibrosis. Cardiac fibroblasts, being so vital in response to cardiac injury, may provide a reasonable starting point for future research.

## Figures and Tables

**Figure 1 cells-11-01316-f001:**
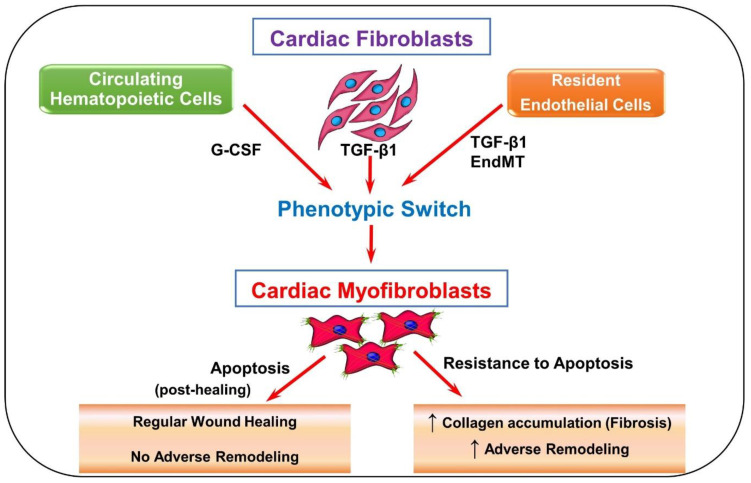
Schematic depicting fibroblast to myofibroblast differentiation. Transforming growth factor-beta 1 (TGF-β1) is the most common pathway involved in the differentiation of fibroblasts into myofibroblasts. Resident fibroblasts, endothelial cells, and circulating hematopoietic cells may have the ability to become myofibroblasts via TGF-β1-induced pathways, granulocyte colony-stimulating factor (G-CSF), and endothelial-to-mesenchymal transition (EndMT). Progenitor stem cells and pericytes are also speculated to contribute to the myofibroblast population. Once myofibroblasts are differentiated, they produce more collagen and various pro remodeling factors required for healing. The expression of these factors at the site of injury promotes repair and remodeling of the injured myocardium. Usually after repair, myofibroblasts undergo apoptosis, decreasing the amount of collagen and pro remodeling factors expressed. When myofibroblasts fail to undergo apoptosis, more collagen and pro remodeling factors are expressed.

**Figure 2 cells-11-01316-f002:**
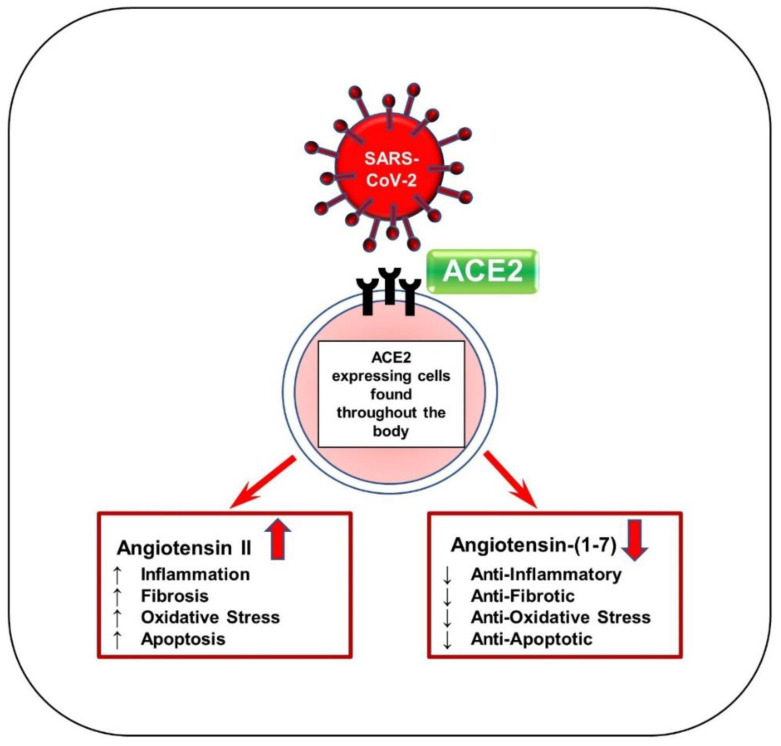
Schematic of SARS-CoV-2 viral entry via ACE2 receptor. SARS-CoV-2 enters the body by first binding to ACE2 receptors by the S protein. ACE2 is responsible for generating Ang-(1–7) from Ang II and acts mainly to reduce blood pressure. This binding can block the ACE2 receptor and reduce ACE2 activity, thereby significantly increasing Ang II levels and drastically decreasing Ang-(1–7) levels. Ang-(1–7) serves as anti-inflammatory, antifibrotic, antioxidative, and antiapoptotic. Ang II increases hypertension and other associated factors such as inflammation, fibrosis, oxidation, and apoptosis.

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
