# Peer review of "CVD and COVID-19: Emerging Roles of Cardiac Fibroblasts and Myofibroblasts"

_cells, 2022, doi:10.3390/cells11081316_

Round 1
Reviewer 1 Report
In their very interesting and well-written review the authors discuss the topic of cardiovascular disease (CVD) and COVID-19 and the involvement of cardiac fibroblasts and myofibroblasts.
The authors introduce in this review key information concerning the role of cardiac myofibroblasts in CVD and COVID-19 complications. Then they give an overview how new factors including controlling ACE2 expression and microRNA regulation are explored as promising treatments for both COVID-19 and CVD. Finally, they conclude that further understanding of this topic may provide insight into the long-term cardiac manifestations of COVID-19 and ways to mitigate any negative effects.
Overall this is a very interesting and timely review. I have only a few minor additions / comments listed below, to broaden the scope of the review considering practical / clinical relevance of these observations, particularly regarding the clinical relevance of these findings.
I have the following comments:
The authors should extend their review by presenting studies which explored non-HLA antibodies in COVID-19 patients. Which clearly demonstrated the relevance and importance of non-HLA antibodies. Especially inclusion of articles with autoantibodies Targeting Angiotensin II receptor 1 (AT1R) could increase the impact of this review. Since there are now few studies highlighting the impact of these antibodies in COVID-19 (PMID: 35264564, PMID: 34788328 and others)
Author Response
Comment 1: “The authors should extend their review by presenting studies which explored non-HLA antibodies in COVID-19 patients. Which clearly demonstrated the relevance and importance of non-HLA antibodies. Especially inclusion of articles with autoantibodies Targeting Angiotensin II receptor 1 (AT1R) could increase the impact of this review. Since there are now few studies highlighting the impact of these antibodies in COVID-19 (PMID: 35264564, PMID: 34788328 and others)”
Authors’ Answer: We appreciate the reviewer’s insightful comments and agree with the suggestion to update the references to include the most recent literature. We have addressed this reviewer’s concerns regarding the above references related to autoantibodies targeting AT1R in lines 203-207 and lines 353-354 and made changes to include those references to improve the manuscript quality, flow, and clarity, while at the same time trying to keep the manuscript within the format of a brief review article.
Reviewer 2 Report
The manuscript shows an interesting review of the scientific literature about molecular mechanisms involved in cardiac fibroblast related to CVDs. However, I think there are several aspects that need further revision:
-The role of cardiomyocytes is completely lacking in the present manuscript. However, a relevant number of references used are based on cardiomyocytes's studies.
-The aim of the review is not clear since there include two different sentences apparently unconnected. Please, explain better the global aim of the study.
-Some of the molecular pathways are repeated or redundant in different subsections. Please, avoid innecesary or duplicated information.
-Section 3.4. Should be titled "Role in the Inmune system of TGF-B1 in Severe COVID-19 cases"
Author Response
Comment 1: “The role of cardiomyocytes is completely lacking in the present manuscript. However, a relevant number of references used are based on cardiomyocytes’s studies.”
Authors’ Answer: We greatly appreciate the reviewer’s insightful critique about cardiomyocytes, which are major cell types that do undergo apoptosis; however, we feel they are slightly out of scope for this review. We decided the focus of this manuscript would be on the cells (fibroblasts and myofibroblasts) responsible for excessive collagen turnover and adverse remodeling of the heart that alters cardiac structure and function. In lines 131-134, we refer to the effect excessive collagen deposition (fibrosis) can have on the communication between cardiomyocytes. Fibrosis can cause a disruption in cardiomyocyte communication and thus heart failure, but cardiac fibroblasts and myofibroblasts are the main cell types responsible for the production of collagen. Therefore, they were our focus.
As for the studies using cardiomyocytes specifically, we have mentioned in the text that the studies involve cardiomyocytes. Additionally, we have added lines 134-135 to further contextualize the role of cardiomyocytes in the adverse remodeling story.
Comment 2: “The aim of the review is not clear since there include two different sentences apparently unconnected. Please, explain better the global aim of the study.”
Authors’ Answer: We appreciate and agree with the reviewer’s comment that the aims of the review could have been clarified. We have adjusted lines 74-78 to better reflect the aims of the review, and to improve the clarity and flow of the paragraph.
Comment 3: “Some of the molecular pathways are repeated or redundant in different subsections. Please, avoid innecesary or duplicated information.”
Authors’ Answer: We have accordingly revised the review to remove sections of redundancy that we have deemed unimportant including a sentence regarding elevated troponin levels in the third paragraph of section 3.2 and a sentence regarding TGF-β1’s role in the first paragraph of section 3.4.
Comment 4: “Section 3.4. Should be titled ‘Role in the Inmune system of TGF-B1 in Severe COVID-19 cases.’”
Authors’ Answer: We agree with this comment and have renamed section 3.4 as “TGF-β1’s Role in the Immune Systems of Severe COVID-19 Patients.”
Reviewer 3 Report
The aim of this review was to expand upon the current literature concerning the role of cardiac fibroblasts and myofibroblasts in adverse remodeling and cardiac fibrosis. Additionally, the authors present emerging evidence concerning COVID-19’s link to cardiovascular disease. The work is well organized and offers an interesting contribution to the area of knowledge.
Author Response
Comment 1: “The aim of this review was to expand upon the current literature concerning the role of cardiac fibroblasts and myofibroblasts in adverse remodeling and cardiac fibrosis. Additionally, the authors present emerging evidence concerning COVID-19’s link to cardiovascular disease. The work is well organized and offers an interesting contribution to the area of knowledge.”
Authors’ Answer: We greatly appreciate the reviewer's time and effort spent reviewing our manuscript and thank them for their comments and scores.
Round 2
Reviewer 2 Report
All my concerns have been propertly assessed by the authors.